# Identification of a Novel SSTR3 Full Agonist for the Treatment of Nonfunctioning Pituitary Adenomas

**DOI:** 10.3390/cancers15133453

**Published:** 2023-06-30

**Authors:** Daniela Modena, Maria Luisa Moras, Giovanni Sandrone, Andrea Stevenazzi, Barbara Vergani, Pooja Dasgupta, Andrea Kliever, Sebastian Gulde, Alessandro Marangelo, Mathias Schillmaier, Raul M. Luque, Stephen Bäuerle, Natalia S. Pellegata, Stefan Schulz, Christian Steinkühler

**Affiliations:** 1Preclinical R&D, Italfarmaco Group, 20092 Cinisello Balsamo, Milan, Italy; 2Institute of Pharmacology and Toxicology, Universitätsklinikum Jena, Friedrich-Schiller-Universität, 07747 Jena, Germany; pooja.dasgupta@med.uni-jena.de (P.D.); andrea.kliewer@med.uni-jena.de (A.K.); stefan.schulz@med.uni-jena.de (S.S.); 3Institute for Diabetes and Cancer, Helmholtz Zentrum München, 85764 Neuherberg, Germanyalessandro.marangelo@helmholtz-muenchen.de (A.M.);; 4Joint Heidelberg-IDC Translational Diabetes Program, Heidelberg University Hospital, 69120 Heidelberg, Germany; 5Department of Biology and Biotechnology “L. Spallanzani”, University of Pavia, 27100 Pavia, Italy; 6Department of Nuclear Medicine, Klinikum Rechts der Isar, School of Medicine, Technical University of Munich, 80333 Munich, Germany; 7Department of Diagnostic and Interventional Radiology, Klinikum Rechts der Isar, School of Medicine, Technical University of Munich, 80333 Munich, Germany; 8Maimonides Biomedical Research Institute of Cordoba (IMIBIC), 14004 Cordoba, Spain; raul.luque@uco.es; 9Department of Cell Biology, Physiology and Immunology, University of Cordoba, 14004 Cordoba, Spain; 10Reina Sofia University Hospital (HURS), 14004 Cordoba, Spain; 11CIBER Physiopathology of Obesity and Nutrition (CIBERobn), 14004 Cordoba, Spain; 12Department of Mathematics, Technical University Munich, 85748 Garching, Germany

**Keywords:** nonfunctioning pituitary adenomas (NFPAs), somatostatin agonists (SSAs), somatostatin receptor 3 (SSTR3), ITF2984

## Abstract

**Simple Summary:**

Therapy for nonfunctioning pituitary adenomas (NFPAs) is a significant unmet medical need since these adenomas are frequently invasive, are difficult to resect completely and often recur. NFPAs express high levels of somatostatin receptor 3 (SSTR3), and SSTR3 agonists could be a promising new treatment, but SSTR3 agonists with sufficient characterization to allow clinical testing are not available. We unexpectedly discovered that ITF2984, a molecule originally developed as a pan-SSTR agonist, is a full agonist of SSTR3 in in vitro assays. A similar full agonism was not observed with Pasireotide another pan-SSTR agonist approved for the treatment of Cushing’s disease. These unexpected findings prompted us to test ITF2984 in an in vivo model that recapitulates the human disease, where ITF2984 showed significant antitumor activity. ITF2984 has completed phase II clinical trials in acromegaly patients, is well tolerated and can be directly tested in NFPA patients, potentially providing a long-sought-after therapeutic option.

**Abstract:**

Somatostatin receptor (SSTR) agonists have been extensively used for treating neuroendocrine tumors. Synthetic therapeutic agonists showing selectivity for SSTR2 (Octreotide) or for SSTR2 and SSTR5 (Pasireotide) have been approved for the treatment of patients with acromegaly and Cushing’s syndrome, as their pituitary tumors highly express SSTR2 or SSTR2/SSTR5, respectively. Nonfunctioning pituitary adenomas (NFPAs), which express high levels of SSTR3 and show only modest response to currently available SSTR agonists, are often invasive and cannot be completely resected, and therefore easily recur. The aim of the present study was the evaluation of ITF2984, a somatostatin analog and full SSTR3 agonist, as a new potential treatment for NFPAs. ITF2984 shows a 10-fold improved affinity for SSTR3 compared to Octreotide or Pasireotide. Molecular modeling and NMR studies indicated that the higher affinity for SSTR3 correlates with a higher stability of a distorted β-I turn in the cyclic peptide backbone. ITF2984 induces receptor internalization and phosphorylation, and triggers G-protein signaling at pharmacologically relevant concentrations. Furthermore, ITF2984 displays antitumor activity that is dependent on SSTR3 expression levels in the MENX (homozygous mutant) NFPA rat model, which closely recapitulates human disease. Therefore, ITF2984 may represent a novel therapeutic option for patients affected by NFPA.

## 1. Introduction

Nonfunctioning pituitary adenomas (NFPAs) are benign adenohypophyseal tumors not associated with clinical evidence of hormonal hypersecretion. NFPAs are mainly gonadotroph pituitary adenomas (GPAs), and account for approximately 35% (14–54%) of all pituitary tumors. Their prevalence is 7–41.3/100,000, the standardized incidence rate is 0.65–2.34/100,000 and the peak occurrence is from the fourth to the eighth decade [1,2]. NFPAs are often diagnosed at the occurrence of clinical signs and symptoms of “mass effects” such as headaches, visual disorders and/or cranial nerve dysfunction caused by compression and lesions extending into the cavernous sinus and the sellar floor [3]. Moreover, some cases are diagnosed incidentally through imaging studies performed for other purposes. Hypopituitarism and hyperprolactinemia, due to the compression of the normal anterior pituitary and to pituitary stalk deviation, respectively, can also be present.

Currently, the standard first-line therapies for most NFPAs are endoscopy or microscopy-assisted transsphenoidal surgery and transcranial surgery, the latter being predominantly used for suprasellar tumors [4]. After surgical treatment, NFPAs often progress, with regrowth rates of 15–66% in NFPA patients treated with surgery alone and 2–28% in those treated with surgery followed by radiotherapy [5,6]. Notably, the systematic use of radiotherapy is limited by its side effects. Therefore, an adjuvant and/or alternative postoperative therapy is a relevant medical need.

Despite their frequency, no standard-of-care drug treatment is currently recommended for NFPAs/GPAs [7,8]. In this context, three main classes of peptides have been studied for the treatment of NFPAs: dopamine receptor 2 (DR2) agonists, somatostatin agonists (SSA) and gonadotropin-releasing hormone (GnRH) analogues. In addition, the use of temozolomide has been introduced in some centers as a therapy for aggressive tumors [9].

The efficacy of DR2 agonists (cabergoline and bromocriptine) correlates with the expression of the receptor, and some studies demonstrated efficacy only when the drugs were administered immediately after surgery [10].

No evidence of GnRH analog efficacy has been demonstrated. Moreover, GnRH agonists exacerbated gonadotropin secretion, with no change in tumor volume or induced pituitary apoplexy when used as therapy for metastatic prostate carcinoma in patients also bearing gonadotroph adenoma [7].

SSAs such as Octreotide (Figure 1A) and Lanreotide (Figure 1D), which bind to somatostatin receptor 2 (SSTR2), and to a lesser extent, to SSTR5 and SSTR3, are effective in the treatment of secreting pituitary adenomas [11,12,13], but are poorly efficacious in NFPAs [14]. Similarly, the panagonist Pasireotide (Figure 1B), which binds to SSTR1, 2, 3, 5, showed only modest efficacy in a recent phase II clinical trial (NCT01283542—Evaluate the Efficacy and Safety of Pasireotide LAR (Long Acting Release) on the Treatment of Patients With Clinically Nonfunctioning Pituitary Adenoma—Passion I), with only 16.7% of patients reaching a tumor size reduction of at least 20% [8,15].

SSTR3 activation by somatostatin (SST) and SSAs induces cytostatic and cytotoxic effects by interfering with mitogenic pathways through the activation of protein tyrosine phosphatases and the subsequent inactivation of Raf1 and MAPK (Mitogen Activated Protein Kinase). In addition, SSTR3 engagement has been proposed to induce apoptosis through p53 and caspase activation. SSTR3 targeting also inhibits endothelial cell proliferation, and consequently, neoangiogenesis [11,16,17,18].

Several studies report that SSTR3 is frequently and strongly expressed in gonadotroph adenomas, while SSTR2 is expressed only in a small number of patients and SSTR5 expression is found only exceptionally [3,8,19,20].

Moreover, besides pituitary adenomas, recent studies suggest an elevated SSTR3 expression in diverse neuroendocrine-related malignancies such as pancreatic tumors [21], pheochromocytomas, paragangliomas [22], lung carcinoids [23] and breast cancer [24].

The development of SSAs that recognize and activate SSTR3 is a potentially promising strategy, due to the fact that (1) NFPAs mainly express SSTR3, which is also maintained after radiotherapy [3] and (2) the response of pituitary adenomas to SSAs depends on the expression of specific SSTR subtypes, as seen for SSTR2 in GH-secreting adenomas [25]. This concept has recently been confirmed experimentally using SSTR3 selective peptides in preclinical models [26].

The aim of this study was the evaluation of ITF2984 (Figure 1C), a novel, cyclic hexapeptide SSA panagonist, as a novel treatment option for NFPAs. ITF2984 behaves as a full agonist on SSTR3, promoting receptor internalization and phosphorylation. This agent showed gender-dependent antitumor activity in the MENX (homozygous mutant) NFPA rat model, which closely resembles the human counterpart [17,27]. Indeed, ITF2984 suppressed tumor growth in female rats having high baseline expression of the *Sstr3* gene, but not in male rats with lower *Sstr3* gene expression.

These data set the rationale for the clinical use of ITF2984 for the treatment of NFPA.

## 2. Materials and Methods

### 2.1. Synthesis of ITF2984

Fmoc-protected amino acid (4 eq.) was dissolved in DMF. HBTU (4 eq.), HOBt (4 eq.) and DIPEA (8 eq.) were added. The reaction mixture was then added to the resin (1 eq.) and stirred at r.t. for 2 h. The resin was filtered and washed with DMF and DCM. The synthesis of Cyclo[Tyr(Bn)-Phe-Pro(4-OCONH(CH2)2NH2)-Tyr-3,8-diMeONal-Lys] (ITF2984) was performed as described in WO2009071460 [28].

### 2.2. Radioligand Binding to Human SSTR

The binding affinity of ITF2984 for human SSTR subtypes was determined in competitive radioligand binding using cell membrane of CHO-K1 cell line expressing hSSTR1, hSSTR2, hSSTR3, hSSTR4, hSSTR5, respectively. GeneBank protein sequences NP_001040.1, NP_001041.1, NP_001042.1, NP_001043.1. SST28 (Bachem, Bubendorf, Switzerland H-4955) were used as reference compounds.

Reaction mix containing SST agonist at increasing concentrations (range 10^−11^–3 × 10^−7^ M, *n* = 2), membrane extracts, radioligand (3-[^125^I] iodotyrosyl^11^ Somatostatin-14 Amersham, IM161, 2000 Ci/mmol was incubated 60 min at 25 °C, filtered and counted for radioactivity with a TopCount^TM^ or MicroBeta^TM^ for 1 min/well after addition of Microscint 20 (Packard, Palo Alto, CA, USA) (see Appendix A for details) [29,30].

Data were analyzed with XLfit (IDBS) software (version 5.3.1.3) using nonlinear regression applied to a sigmoidal dose–response model. Agonist activity of test compounds is expressed as a percentage of the activity of the reference agonist at its EC_100_ concentration.

### 2.3. SSTR Internalization

#### 2.3.1. HEK293 Cell Line

HEK293 cells obtained from DSMZ (Braunschweig, Germany) were transfected with plasmid encoding murine HA-tagged SSTR2, SSTR3 or SSTR5 receptors [31,32,33].

HEK293 cells stably expressing HA-tagged hSSTR cultures were preincubated with anti-HA antibody for 2 h at 4 °C and then exposed to 1 µM agonist for 30 min at 37 °C before fixing and further labeling with Alexa488-conjugated secondary antibody. After mounting with Roti^®^-MountFluorCare DAPI, cultures were examined using a Zeiss LSM510 META laser scanning confocal microscope (Zeiss, Jena, Germany) (see Appendix A for details).

#### 2.3.2. U2OS Human Cell Lines Transfected with Human Receptors (SSTR2-tGFP, SSTR3-tGFP, SSTR5-tGFP)

Compounds were tested at five concentrations (10^−5^, 10^−6^, 10^−7^, 10^−8^, 10^−9^ M) in comparison to untreated cells. A concentration of 10^−6^ M SST28 (Sigma-Aldrich S6135) was included as positive control.

SST agonists were added to U2OS recombinant cell lines in OptiMeM medium (Life technologies 51985-034, Monza, Italy) for 3 h (SSTR2, SSTR3) or 7 h (SSTR5) (n = 3). After formaldehyde fixation, nuclei were stained using DAPI (2 µg/mL), and the fluorescence was measured with BD Pathway 855 High-Content Bioimager from Becton Dickinson. The receptor internalization was calculated using AttoVision 1.6 Software. Approximately 500 cells per field were analyzed.

Both Excel 2003 and Sigmaplot 9.0 were used for data management.

Agonist activity of test compounds was calculated relatively to positive control (SST28 10^−6^ M) and is shown as a percentage of activity.

### 2.4. SSTR Phosphorylation

Western blot analysis was performed as previously described [31,32,33]. Briefly, HEK293 cells stably expressing HA-SSTR3, HA-SSTR2 or HA-SSTR5 were either treated with 10 µM SST14, Octreotide, Pasireotide or ITF2984 with concentrations ranging from 10^−5^ M to 10^−12^ M for 5 min at 37 °C. Cells were lysed and immunoblotted with phosphosite-specific antibodies: pS337/pT341-SST3 (7TM0357A), pS341/pS343-SST2 (7TM0356A), pT333-SST5 (7TM0359A) (7TM Antibodies GmbH, Jena, Germany) (1:1000) at 4 °C overnight, followed by HRP-linked secondary antibody for 2 h with phosphorylation-independent SSTR antibodies non-phospho-SST3 (7TM0357N), non-phospho-SST2 (7TM0356N), non-phospho-SST5 (7TM0359N) or anti-HA antibodies (7TM000HA) (7TM Antibodies GmbH, Jena, Germany) to confirm equal loading of the gel (see Appendix A for details).

### 2.5. Membrane Potential Assay

HEK293 cells stably transfected with either HA-tagged SSTR2, SSTR3 or SSTR5 and GFP-conjugated GIRK2 channel plasmids (Origene, Rockville, MD, USA) were seeded in 96-well plates and allowed to grow at 37 °C and 5% CO_2_ for 48 h. After washing, 90 µL of the HBSS/HEPES buffer solution and an equal volume of the membrane potential dye (FLIPR Membrane Potential kit BLUE, Molecular Devices, San Jose, CA, USA) was added to each well and cells were incubated for 45 min at 37 °C. Fluorescence measurements were performed in a FlexStation 3 microplate reader (Molecular Devices) at 37 °C with Ex = 530 nm and Em = 565 nm. Baseline readings were taken every 1.8 s for 1 min. After 60 s, a volume of 20 µL of the test compounds (10×) or vehicle was injected into each well containing cells incubated with dye. The change in fluorescence of the dye was recorded for 240 s using SoftMax Pro software (see Appendix A for details) [34,35].

### 2.6. In Vivo Efficacy Experiments

Rats were maintained in agreement with general husbandry rules approved by the Helmholtz Zentrum München. All experimental procedures were conducted in accordance with committed guidelines as approved by local government (GV-Solas; Felasa; TierschG). In vivo studies were approved by the government of Upper Bavaria, Germany (rat studies: Az. 55.2.1.54-2532-39-13 and 55.2-2532.Vet_02-18-102).

MENX rats were injected s.c. with ITF2984 at the dose of 12.5 mg/kg body weight (bw) or with placebo (PBS) (control group) every 14 days.

#### 2.6.1. Magnetic Resonance Imaging (MRI)

Tumor monitoring in rats was conducted using MRI at day 0 (pretreatment) and every 14 days post-treatment (days 14, 28, 42, 56). MRI was performed on a 7T preclinical scanner (Bruker BioSpin MRI GmbH, Ettlingen, Germany) using previously published procedures suitable to visualize the rat pituitary tumors. In the T2 weighted datasets, regions of interest (ROIs) were manually segmented around the adenomas in every slice where they appeared; tumor volumes were finally calculated from the data (area of particular ROIs and slice thickness) using an implemented algorithm in Osirix/Horos (Pixmeo SARL/Horos Project) as previously described [36].

#### 2.6.2. RNA Extraction and Quantitative RT-PCR

RNA was extracted using RNeasy Mini Kit (Qiagen, Cluj-Napoca, Romania) following the manufacturer’s instructions. Total RNA concentration and purity was assessed using Nanodrop 2000 spectrophotometer (Thermo Scientific, Waltham, MA, USA). Total RNA was retrotranscribed using random hexamer primers and the cDNA First Strand Synthesis kit (Thermo Scientific). Details regarding the quantitative PCR (qPCR) procedure used to determine the absolute expression levels of the different somatostatin receptor genes (*Sstr1-5*) have been previously reported [37]. Specific sets of primers for these genes have been previously validated and reported [37]. To control for the amount of RNA and for the efficiency of the retrotranscription reaction, mRNA copy numbers of the different transcripts analyzed were adjusted by β-actin (ACTB) expression (used as housekeeping gene).

#### 2.6.3. Immunohistochemistry

From the paraffin blocks sections 4 µm in length were prepared and floated onto positively-charged slides. Immunostaining for SSTRs was performed by an indirect peroxidase labeling method according to previously published protocols [31,38]. The anti-SST antibodies used for the study were SST1: E4317, anti-human/rat/mouse, rabbit polyclonal, affinity purified, concentration 1 µg/mL; SST2: UMB-1, anti-human/rat/mouse, rabbit monoclonal, cell culture supernatant, dilution: 1:10; SST3: 1308, anti-rat, rabbit polyclonal, affinity purified, concentration 1 μg/mL; SST5: 6003, anti-rat, rabbit polyclonal, dilution: 1:2000 [31,38].

Immunohistochemistry for Ki-67 (BD Pharmingen, San Diego, CA, USA 556,003, dilution: 1:1000) and Cleaved caspase 3 (Cc3, Cell Signaling, Danvers, MA, USA, 9664, dilution: 1:150) was performed using a Bond RXm system (Leica, Wetzlar, Germany, all reagents from Leica). Briefly, slides were deparaffinized and pretreated with Epitope retrieval solution 1 (corresponding to citrate buffer pH6) for 20 min. Tissues were incubated with the primary antibody for 15 min at room temperature. Antibody binding was detected with a polymer refine detection kit used without postprimary reagent for the Cc3 antibody and visualized with DAB as a dark-brown precipitate. Counterstaining was performed with hematoxyline.

Ki67-positive tumor cells per 100,000 µm^2^ were counted under the microscope in 3 independent areas for each tumor sample.

#### 2.6.4. Statistical Analysis

Linear mixed-effects (LME) models were applied for longitudinal analysis of tumor volume growth, with absolute tumor volume at day 0 used for scaling subsequent measurements of each individual, as previously reported [36]. Relative volumes were transformed by natural logarithms for use as model outcomes in order to meet the normal distributional assumptions. Linear and quadratic growth predictors and interactions were considered for significance testing, performed by the F-test, with results presented as the mean ± standard error of the mean (SEM). Statistical significance between two series of data was determined by one-way ANOVA. A *p* value < 0.05 was considered statistically significant.

## 3. Results

### 3.1. Molecular Modeling

ITF2984 (Figure 1C), a novel SST panagonist cyclic hexapeptide, was discovered in a medicinal chemistry program aimed at identifying panagonists with improved properties versus first-generation SSAs. This compound showed high binding affinity for SSTR1, 2, 3 and 5, with IC_50_ values in the nanomolar range for all receptors (Table 1). When compared with Octreotide, ITF2984 showed higher affinity for SSTR1, SSTR3, and SSTR5 and lower affinity for SSTR2, whereas relative to Pasireotide, it exhibited higher affinity for human SSTR1, SSTR2 and SSTR3 (Table 1). Noticeably, the IC_50_ values of ITF2984 on SSTR3 were about one order of magnitude lower than those obtained with either Octreotide or Pasireotide.

A molecular modeling approach was used to rationalize the structural basis for the higher affinity of ITF2984 for SSTR3. To this end, the SSTR structure from GPCRdb [39], an open-access repository of G-coupled protein receptor structures, was used. Models of receptors SSTR1-4 are mainly based on the κ-opioid receptor (or KOP, sequence similarity 60–66%, pdb codes 6B73 and 6VI4 for active and inactive conformer, respectively), whereas SSTR5 is more similar to the δ-opioid receptor. The relevance of using the opioid receptors in this modeling exercise was experimentally confirmed (Table 2) by ITF2984 and Pasireotide single-dose binding assays on this particular GPCR family, where both hexapeptides were able to fully replace agonists at 10^−5^ M.

During manuscript preparation, several structures of somatostatin receptor type 2 (SSTR2) were published and became accessible on the protein databank. Although SSTR4 was also investigated, the main focus was on SSTR2, with structures obtained using (1) Cryo-EM techniques [40,41,42,43,44,45], where the receptor is bound to the somatostatin-14, agonist, Octreotide (Sandostatin) and Lanreotide [PDB codes (7XAU, 7XAV, 7XAT) [40], (7XMT, 7XMR, 7XMS) [45], (7Y27, 7Y26, 7Y24) [42], (7WIG, 7WIC) [41], (7WJ5) [43], (7T10, 7UL5, 7T11) [44]; or (2) X-ray diffraction pattern refinement, where SSTR2 interacts with small molecules (agonist L-054,522, antagonist CYN 154806, PDB Codes 7XNA, 7XN9) [45].

Figure 2A reports the interaction network between the key dyad Trp -Lys and receptor residues lying in a sphere with a radius equal to 5.0 Å around the residue pair. SRIF-14 (pdb code 7T10, Cryo-EM, resolution 2.5 Å) shows that residue Lys9 interacts with Asp122 (salt bridge) [46], and Tyr302 (triggering a charged assisted hydrogen bond) [47] with Val298, whereas the Trp8 side chain is buried in a hydrophobic pocket formed by Phe208 and Phe272, where noncovalent π-π interactions [48] are prevalent. The backbone torsions (Appendix A) related to tryptophan and lysine indicate that the ligand shows [49] a structured region: a β-II’ turn motif [50,51], centered in the dyad. It should be noted that the two complexes confirm (Appendix A) the particular secondary structure detail (PDB structure 7Y27 and 7WJ5); one exhibits three out of four compatible dihedral angles (PDB code: 7XMR), whereas the last conformers (PDB code 7WIC and 7XAT) show a different turn (β-II and β-I, respectively). All complexes involving SSTR2 and Octreotide (7Y26, 7Y24, 7XAU, 7T11) and Lanreotide (7XAV) clearly indicate that the key dyad D-Trp-Lys exhibits a β-II’ turn, a feature already detected in early NMR studies of Octreotide [52]. The presence of such a structural motif in all complexes between SSTR2 and related selective and highly potent cyclic somatostatin octapeptide analogues, as well as the presence of a β-II’ turn in several structures involving SRIF-14, suggests that the secondary structure motif could be regarded as a pharmacophore feature connected to the activity of somatostatin (and analogues) on receptor type 2.

Molecular dynamics simulations (MDs) [53] can reinforce the hypothesis regarding the role of the β-II’ turn as a determinant for affinity on SSTR2 and allow one to explore the behavior of ITF2984 and/or Pasireotide on the type 2 receptor.

The structure of the 7-helix receptor interacting with SRIF-14 (pdb code 7T10) was extracted from the large complex in which SSTR2 is linked to the Gi3 protein, assuming that the removal of the associated protein does not significantly perturb receptor–ligand dynamics. The peptide–receptor ensemble was then buried in a membrane automatically built by using 1-palmitoyl-2-oleoylphosphatidylcholine or POPC, as proposed by System Builder of Schrodinger Desmond module [53]. The size of the orthorhombic cell was defined in terms of the 7-helix receptor size, whereas the outermost hydrophilic region of the membrane was automatically solvated. Simulations were run for 300 ns (NPT ensemble, 300 K, 1 bar), and the first 100 ns were considered as equilibration and discarded. An analogous model was derived from the Octreotide-SSTR2 complex (PDB code 7T11, originally including the Gi3 protein, removed in the preliminary model preparation), soaked in the POPC matrix and solvated in the outermost hydrophilic region. Finally, ITF2984 and Pasireotide agonists were docked in both conformers of SSTR2 derived from complexes of receptor type 2 and somatostatin 14 and Octreotide (Glide, Schrodinger suite [54]). The highest-scoring pose turned out to be related to SSTR2 originally bound to SRIF-14 in both cases, allowing for the last step of the model preparation, (membrane setup) following the protocol adopted for SIRF14 and Octreotide. Figure 2B reports the average value of torsions for the key dyad X-Lys (X = Trp, D-Trp, 3,5 di-methoxy-D-2-naphtylalanine for SRIF-14, Octreotide/Pasireotide and ITF2984, respectively). The average values of backbone torsions (Appendix A) exhibited by the X-Lys substructure of somatostatin-14 and Octreotide during production (from 100 ns to 300 ns) confirm that β-II’ is stable and represents the secondary structure of the bound conformer. Pasireotide shows a different motif (β-II), undetected in experimental complexes, whereas the dyad in ITF2984 apparently exhibits two privileged structures (β-II’ and β-I), due to the high standard deviation of ϕ(i + 1) torsion, probably induced by a suboptimal fit of the side chain of unnatural residue X.

The investigation of ligands in SSTR3 was accomplished using an SSTR3 homology model available in the GPCRDB data bank, and the starting binding geometry of all ligands was derived from a preliminary docking simulation. All complexes were then buried in the POPC membrane model, then solvated in the outermost part of the orthorhombic cell, following the previously adopted workflow. All somatostatin analogues (Appendix A) apparently adopted a binding conformer having the same turn found in SSTR2, but SRIF-14 preferred a β-I turn, a feature partially exhibited by ITF2984 in the related complex, although with a distortion. It could be speculated that the β-II’ turn is an undesired ligand feature on SSTR3, explaining the reduced potency of Octreotide on receptor type 3. The β-II conformer also leads to a reduced activity of Pasireotide on both GPCRs, whereas the fluctuating behavior between the β-II’ and β-I turns, detected in the X-Lys dyad backbone of ITF2984, could explain the partial preservation of binding potency on SSTR2 and the remarkable activity exhibited on SSTR3.

We further characterized the biological activity of ITF2984: This molecule potently inhibited GH release from rat anterior pituitary primary cell cultures with no statistical difference relative to Pasireotide (Appendix A) [55,56]. 

### 3.2. Internalization of SSTR2, SSTR3, SSTR5

To highlight the mechanistic differences between ITF2984 and other SSTR agonists, we investigated SSTR internalization using two different experimental models: (i) HEK293 cells transfected with human SSTRs, and (ii) U2OS cells transfected with human GFP-tagged receptors (SSTR2-tGFP, SSTR3-tGFP, SSTR5-tGFP). SRIF14 and SST28 were included as reference compounds in the first and second experiments, respectively.

In transfected HEK293 cells, ITF2984 induced a strong internalization of SSTR3, which was comparable to that induced by SRIF14 and significantly higher than those induced by Octreotide or Pasireotide (Figure 3A). In contrast, only a partial internalization of SSTR2 and SSTR5 was observed.

In U2OS cells, all compounds increased SSTR2-tGFP internalization in a dose-dependent manner when compared to the untreated control, with Octreotide being the most potent internalization inducer, followed by Pasireotide and ITF2984. In SSTR5-tGFP-transfected cells, the increased receptor internalization due to the treatment with SSAs was similar to the effect of SST, albeit lower (about fifty percent of SST28). Lastly, in SSTR3-tGFP-transfected cells, ITF2984 was the most potent inducer of receptor internalization, showing an increment comparable to SST28 at 1 µM. At the same concentration, the effect of Octreotide and Pasireotide was significantly less pronounced (Table 1, Figure 3B).

We conclude that ITF2984 induces SSTR3 internalization more efficiently than Pasireotide or Octreotide in two different cell lines.

### 3.3. SSA-Induced Activation of Human SSTR2, SSTR3 and SSTR5 Receptors Using Phosphosite-Specific Antibodies

Next, the activation of human SSTRs induced by SSAs was tested by Western blot using phosphosite-specific antibodies directed against the following residues: S341/pS343 for SSTR2; S337/T341 for SSTR3; T333 for SSTR5 [31]. ITF2984 induced the full phosphorylation of SSTR3 in a dose-dependent manner (Figure 4A left) and in a pharmacologically relevant, nanomolar dose range, whereas Pasireotide only induced a weak effect in a suprapharmacologic, micromolar dose range. In SSTR2, both cyclohexapeptides induced a selective phosphorylation of S341/S343, whereas SRIF14 and Octreotide led to the full phosphorylation of SSTR2 (Figure 4B left). Finally, the SSTR5 pT333 phosphosite was only partially affected by all tested compounds, with Pasireotide showing the most pronounced effect (Figure 4C left).

### 3.4. Agonist-Mediated G Protein Signaling of SSTR3 in HEK293 Cells and in AtT20 SSTR3-Transfected Cells

Somatostatin receptor signaling is known to activate G protein-coupled inwardly rectifying potassium (GIRK) channels [31]. Therefore, G protein signaling mediated by SSTR agonists was studied using a GIRK-based fluorescence membrane potential assay both in HEK293 cells transfected with hSSTR3, and in AtT20 wild-type as well as hSSTR3-transfected cells. Stimulation of HEK293 cells stably expressing human SSTR3 receptor with SRIF14, Octreotide, Pasireotide and ITF2984 resulted in a dose-dependent reduction in the fluorescent signal of the FMP dye, with ITF2984 being more potent than Octreotide or Pasireotide (Figure 4A–C right). To better characterize ITF2984, the GIRK channel activation was also applied to SSTR2 and SSTR5. In HEK293-GIRK2-GFP-HA-hSST2, the most potent agonist was Octreotide followed by Pasireotide and ITF2984, while in HEK293-GIRK2-GFP-HA-hSST5, the highest activation was induced by Pasireotide. Results of agonist-mediated G protein signaling of SSTR2, SSTR3, SSTR5 in HEK293 cells are summarized in Table 1. Additionally, G protein signaling mediated by Octreotide and ITF2984 was analyzed in mouse corticotroph tumor AtT-20 cells, which endogenously express GIRK1/2 channels as well as SSTR2 and SSTR5 receptors. The exogenous expression of the SSTR3 receptor resulted in a leftward shift of the ITF2984-mediated dose–response curve, but not of the dose–response curve obtained with Octreotide, suggesting the engagement of SSTR3 by ITF2984 but not by Octreotide under these conditions (Figure 5). These data agree with those obtained with the hSSTR3-transfected HEK293 cell line.

### 3.5. Efficacy of ITF2984 In Vivo

#### 3.5.1. Efficacy of ITF2984 against Endogenous NFPAs In Vivo

Encouraged by the in vitro findings, we decided to study the in vivo antitumor activity of ITF2984 in the MENX (homozygous mutant) rat model, which is the only spontaneous, endogenous model where NFPAs develop with complete penetrance [17,27]. NFPAs that develop in this model were found to recapitulate both the histopathology [27] and the SSTR expression pattern of their human counterparts, showing high SSTR3 expression [36]. Interestingly, SSTR3 levels in MENX rat pituitary tumors were found to be gender-specific, with higher expression observed in females. This pattern may also extend to human NFPAs. Rats of both genders were treated for 56 days with ITF2984 or placebo, and tumor growth was monitored longitudinally using high-resolution magnetic resonance imaging (MRI).

Male rats treated with ITF2984 and control group members showed a rapid increase in relative tumor volume, (Figure 6A), exhibiting a logarithmic value best fitted by a linear mixed-effects model (LME) with quadratic time effects (Appendix A). In contrast, female rats treated with ITF2984 showed only a slight increase in tumor volume during treatment (Figure 6A, Appendix A). The difference in time slopes between sexes for the ITF2984-treated rats was significant (*p*-value = 0.0251) (Appendix A). This indicates that female mutant rats responded significantly better to ITF2984 when compared to males. Remarkably, although two female rats had already relatively large tumors at the beginning of the study, ITF2984 kept tumor growth under control, and these two animals showed no signs of discomfort at the end of treatment. When combining both sexes, the overall reduction in tumor growth in ITF2984-treated rats (used as proxy for drug response) versus that in the control group, as assessed by the best-fitted LME, was modest (Appendix A). However, when sexes were analyzed separately, drug-treated female rats showed a suppression of tumor growth when compared to their placebo-treated counterpart, indicating that the former group responded to ITF2984 (Appendix A).

#### 3.5.2. Effect of ITF2984 on NFPA Proliferation Rates

At the end of the treatment, pituitary tissues were collected for ex vivo analyses. Staining for Ki67 was performed on all tumors of rats treated with either placebo or ITF2984, and the number of Ki67-positive cells per 100,000 mm^2^ was counted. Placebo-treated NFPAs (control) showed an average of 375 (males) and 312 (females) Ki67-positive cells per area, as reported [36] (Figure 6B). Not surprisingly, in the control group, there was a positive trend between the number of Ki67-positive cells and absolute tumor volume in males and females [36]. In tumors of rats treated with ITF2984, the number of Ki67-positive cells dropped to an average of 250 (−33.5%) for male and to 134 (−57%) for female rats (Figure 6B and Appendix A). The difference in the number of Ki67-positive cells between placebo- and ITF2984-treated female rats is significant (*p* = 0.031). Changes in tumor cell proliferation correlate with the changes in tumor volume determined by MRI: ITF2984 suppressed tumor growth more effectively in females than in males, and this went together with a stronger reduction in NFPA cell proliferation in the former.

#### 3.5.3. Expression of SSTRs in Rat NFPAs

The expression level of the various *Sstr* genes was assessed in rat tumors at the end of treatment by measuring the copy number for each transcript (absolute quantification) via quantitative RT-PCR. In the placebo control group, females had a statistically significantly higher amount of *Sstr3* mRNA (3-fold) than male rats, as previously reported [36], while other receptors did not differ between the two groups (Appendix A). Tumors from the two animal groups were also stained with antibodies against SSTR1, 2, 3 and 5 using immunohistochemistry (IHC), and the results confirmed the high expression of Sstr3 in female rats [36]. ITF2984 administration led to a downregulation of *Sstr5* expression in both sexes, and to an increase in *Sstr3* mRNAs, but only in females (Figure 6C and Appendix A).

## 4. Discussion

We herein report the characterization of ITF2984, a novel pan-SSTR agonist. While structurally analogous to other SSTR panagonists, ITF2984 has a higher affinity for SSTR3 as compared to known molecules. Molecular modeling revealed that a higher β-I turn probability in ITF2984 correlated with higher SSTR3 affinity, thus providing a structural rationale for the unique selectivity pattern of this molecule.

Unlike Pasireotide or Octreotide, ITF2984 induced SSTR3 internalization and phosphorylation, as well as GIRK activation in a pharmacologically relevant concentration range. Based on these data, ITF2984 can be considered a full agonist of the SSTR3 receptor. ITF2984 properties along with those of Octreotide and Pasireotide are summarized in Table 1.

High SSTR3 expression is found in several malignancies of the neuroendocrine lineage. To preclinically explore the therapeutic potential of ITF2984, we decided to focus on NFPAs, which constitute a significant unmet medical need. The MENX (homozygous mutant) rat model, is a spontaneous, endogenous NFPA model with a 100% penetrance, that closely resembles human disease. Adenomas developing in this model have a gender-specific SSTR3 expression pattern, with high expression levels of the receptor in females. Consistent with its receptor affinity profile and with the in vitro data, ITF2984 showed selective antitumor activity in female—but not in male—rats. This antitumor activity went along with a decrease in the proliferation index (Ki67 positivity) of ITF2984-treated tumors. In addition, ITF2984 selectively induced SSTR3 mRNA expression in female rats, suggesting a compensatory upregulation. We conclude that the data are in line with an in vivo engagement of SSTR3 and a predominantly SSTR3-driven antitumor activity of ITF2984 in this model and provide an in vivo proof of concept for the potential clinical use of ITF2984 in NFPAs and other SSTR3-driven diseases. Since ITF2984 has completed GLP toxicology studies as well as two phase I and one phase II clinical studies (the last of which was in acromegalic patients), clinical testing of this drug for the treatment of NFPA patients should be very straightforward.

Recently, the activities of Pasireotide and Octreotide in the MENX model were reported, and it is instructive to compare those data with the results on ITF2984 reported herein. In the published report [38], Pasireotide was shown to have a higher antitumor activity than Octreotide, which was evident on both female and male rats, with a trend towards a higher activity in females. The enhanced activity was ascribed to an involvement of the SSTR3 receptor. We notice that these data differ from our observations using ITF2984. In fact, ITF2984 showed a significant radiologic tumor response and a compensatory upregulation of SSTR3 mRNA only in female rats, while no significant antitumor effect was observed in male rats with lower SSTR3 levels. We interpret these data in terms of a more selective activity of ITF2984, predominantly, if not exclusively, mediated by SSTR3 engagement under the tested experimental conditions. A difficulty in this comparison emerges from the use of different slow-release formulations for Pasireotide and ITF2984, possibly giving rise to different exposures to the drugs. More accurate head-to-head comparisons at equivalent exposures will have to be carried out to further dissect differences between the two compounds.

One important aspect relates to safety margins. Hyperglycemia is a common side effect of SSTR panagonists such as Pasireotide. In our clinical studies, we also observed increased glycemia at high doses of ITF2984 (manuscript in preparation). What remains to be ascertained clinically is whether the higher agonistic activity on SSTR3 may allow us to define a dose at which full receptor engagement is observed in the absence of significant hyperglycemia.

There are several tumor indications beyond NFPAs that could be targeted with ITF2984. The efficient internalization of ITF2984 upon binding to SSTR3 suggests the possible development of conjugates with potent toxins that could be delivered to SSTR3-expressing cells via ITF2984.

A growing interest toward SSTR3 is also emerging in the field of ciliopathies [57]. Following the first discovery of SSTR3 localization in neuronal cilia [38], many studies have been performed to identify the role of SSTRs in this cell compartment. In the last few years, an important role has been progressively attributed to SSTR3 receptor signaling in neuronal cilia [58]. Recent investigations describe the localization of SSTR3 nearly exclusively to cilia of excitatory neurons, suggesting that pharmacological bidirectional manipulation of this receptor signaling could modulate excitatory synaptic inputs onto these neurons [59]. SSTR3 agonists significantly modulate excitatory synaptic properties, perturbating neuron excitatory–inhibitory balance (E/I) [59], a parameter frequently altered in many brain disorders, including autism spectrum disorder (ASD) [60]. Finally, the involvement of SSTR3 in the signaling necessary for new object recognition memory has been reported [61]. Based on the above considerations, the availability of an SSTR3 full agonist could be promising for the treatment of neurodegenerative pathologies.

## 5. Conclusions

ITF2984 behaves as a full agonist of SSTR3 in vitro, promoting receptor internalization and phosphorylation as well as GIRK activation. This agent showed SSTR3-dependent antitumor activity in the MENX (homozygous mutant) rat model of NFPAs. ITF2984 has completed all preclinical safety studies and was tested in two phase I clinical trials in normal healthy volunteers and in a phase II clinical trial in acromegaly patients, showing efficacy at tolerated doses [55,56]. These data point to a potential use of ITF2984 in NFPA patients or in other SSTR3-dependent diseases.

## Figures and Tables

**Figure 1 cancers-15-03453-f001:**
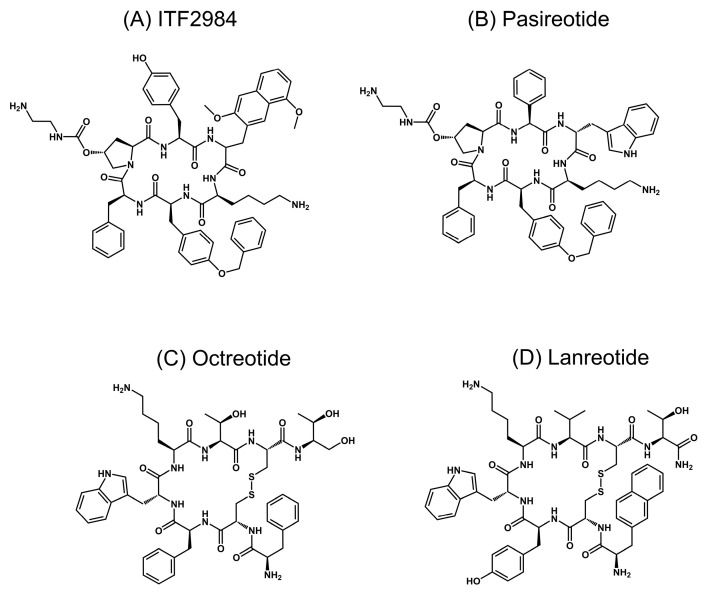
Structures of somatostatin analogues evaluated in the present study. ITF2984 (**A**) and Pasireotide (**B**) are backbone-cyclized hexapeptides, with unnatural amino acids in the most probable active substructure. Approved somatostatin agonists Octreotide (**C**) and Lanreotide (**D**) are cyclic octapeptides conformationally restricted by the disulfide bridge. All molecules share the same key dyad D-Trp-Lys, except ITF2984, where D-Trp is replaced by 3,5 dimethoxy D-2-naphtylalanine (3,5 diMeO-D-2Nal).

**Figure 2 cancers-15-03453-f002:**
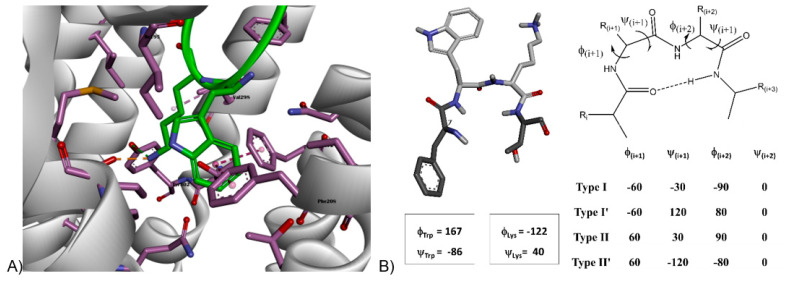
Somatostatin in SSTR2. (**A**) Somatostatin-14 (green) in SSTR2, Cryo-EM structure, resolution 2.5 Å (PDB structure 7T10), detail of binding site hosting the key dyad Trp8-Lys9 (receptor residues close to analogue in purple color). (**B**) Detail of Phe-Trp-Lys-Thr motif of SRIF-14 in best-scored pose (**left**) and generic sketch summarizing reference values of backbone torsions of I, I’, II and II’ β-turns (**right**). Measured dihedral angles ϕ and ψ of Trp-Lys dyad are in qualitative agreement with ideal turn type II’.

**Figure 3 cancers-15-03453-f003:**
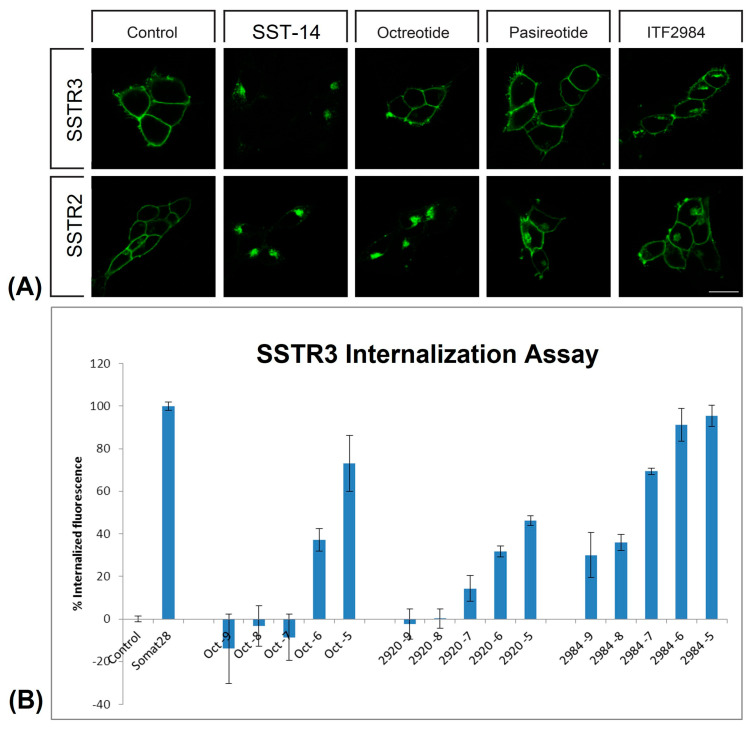
ITF2984-induced internalization of SSTR3 in HEK293 (**A**) and U2OS (**B**) hSSTR-transfected cells. (**A**) HEK293 cells stably expressing wild-type HA-SSTR2 or HA-SSTR3 were preincubated with anti-HA antibody for 2 h at 4 °C. Afterwards, cells were treated with either 1 µM SST-14 (SRIF-14), Octreotide, Pasireotide or ITF2984 for 30 min at 37 °C. After fixation, the cells were incubated with Alexa488-conjugated secondary antibody and examined by confocal microscopy. Shown images are representative of three independent experiments. Scale bar, 20 µm. (**B**) Internalization of SSTR3 in U2OS cell line transfected with human receptor (SSTR3-tGFP) following treatments with SST-28, Pasireotide, ITF2984 and Octreotide. Compounds were tested at five different concentrations (10^−5^, 10^−6^, 10^−7^, 10^−8^, 10^−9^ M) in comparison to untreated cells (Control). A concentration of 10^−6^ M SST28 was included as positive control. Approximately 500 cells per field were analyzed. Data obtained on SSTR2-tGFP- and SSTR5-tGFP-transfected U2OS cells are shown in Table 1.

**Figure 4 cancers-15-03453-f004:**
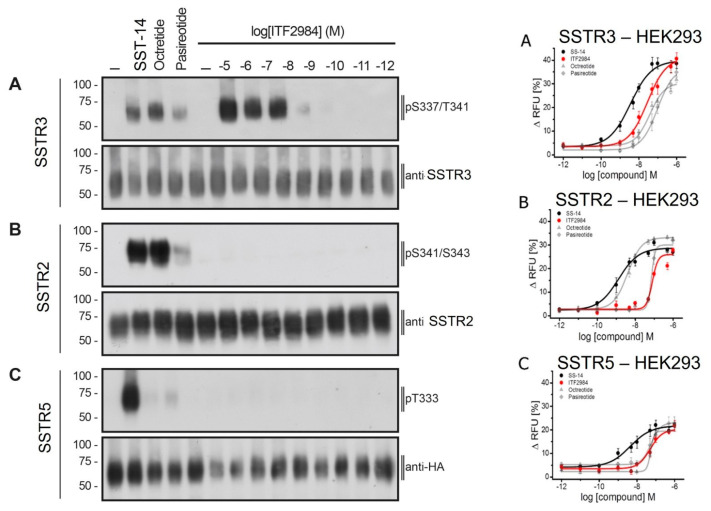
SSTR phosphorylation (**left panel**) and agonist-mediated G protein signaling of SSTRs in HEK293 cells (**right panel**). (**Left panel**) ITF2984-selective SSTR3 phosphorylation in HEK293 cells. HEK293 cells stably expressing (**A**) HA-SSTR3, (**B**) SSTR2 or (**C**) HA-SSTR5 were either treated with 10 µM SST-14, Octreotide, Pasireotide or ITF2984 with concentrations ranging from 10^−5^ M to 10^−12^ M for 5 min at 37 °C. Cells were lysed and immunoblotted with the indicated phosphositespecific antibodies. Blots were stripped and reprobed with the phosphorylation-independent anti-HA-tag or UMB antibody to confirm equal loading of the gels. Blots are representative of three independent experiments. The positions of molecular mass markers are indicated on the left (in kDa). (**Right panel**) Agonist-mediated G protein signaling of SSTR3 in HEK293 cells. The ability of SST14, Octreotide, Pasireotide and ITF2984 to activate GIRK2 channels via SSTR3 (**A**), SSTR2 (**B**) or SSTR5 (**C**) was tested using a fluorescence membrane potential assay. The concentrations used are indicated. Data points represent mean ± S.E.M. The uncropped blots are shown in Appendix A.

**Figure 5 cancers-15-03453-f005:**
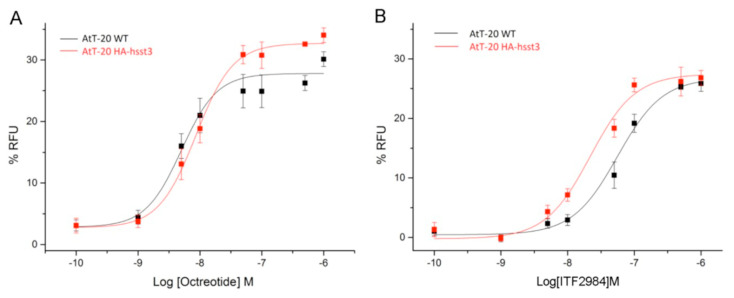
Analysis of G protein signaling in mouse AtT-20 cells using a fluorescence-based membrane potential assay. The ability of Octreotide (**A**) and ITF2984 (**B**) to activate endogenous GIRK channels in wild-type AtT-20 cells via the endogenously expressed SSTR2 and SSTR5 receptors (black) or in exogenously expressed human SSTR3 receptors (red) was tested. Expression of SSTR3 resulted in a leftward shift of the dose–response curve.

**Figure 6 cancers-15-03453-f006:**
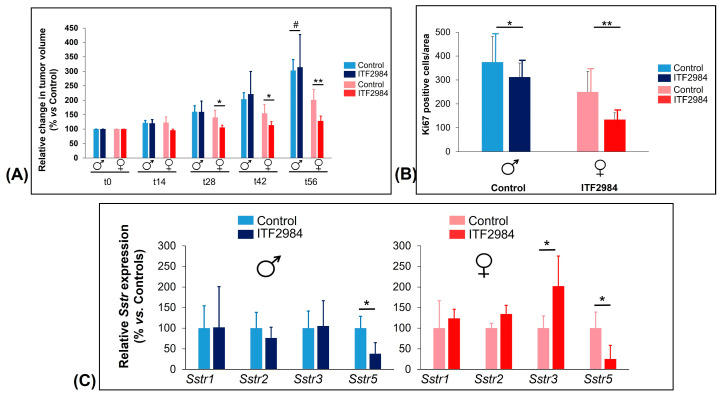
Changes in tumor volume in rats treated s.c. with ITF2984 or placebo (**A**), proliferation rate evaluation (number of Ki67-positive cells) (**B**), *Sstr* gene expression in rat NFPAs (**C**) at the end of treatment. (**A**) Changes in tumor volume in rats treated s.c. with ITF2984 or placebo. MENX-affected rats at the age of 5.5 months were injected with ITF2984 1× every 14 days at 12.5 mg/kg body weight. MRI was performed every 14 days and the tumor volume was normalized against the volume at day 0. Male and female rat tumors are shown separately. Data presented are the mean ± SEM. #, not significant; *, *p*-value < 0.05; **, *p*-value < 0.001. (**B**) Proliferation of NFPAs following treatment. Number of Ki67-positive cells per 100,000 µm^2^ in tumors of rats belonging to the 2 groups (ITF2984-treated and control) and both sexes. Shown is the mean ± SEM. *, *p*-value < 0.05; **, *p*-value < 0.001. (**C**) *Sstr* gene expression in rat NFPAs. Relative expression of the *Sstr*1,2,3,5 genes in the ITF2984 treatment group compared to the control group, arbitrarily set to 100%. Shown is the average ± SEM. *, *p*-value < 0.05. ♂, ♀ are male and female gender symbols, respectively.

**Table 1 cancers-15-03453-t001:** Profile of first (Octreotide)- and second (Pasireotide and ITF2984)-generation SSAs, emerging in the present study. Binding to SSTR, induction of SSTR internalization, induction of SSTR phosphorylation, induction of SSTR G-protein signaling—GIRK2 channel activation in HEK transfected cells and in wild-type (WT)t and transfected AtT20 cells. Qualitative description in terms of color code: excellent (green); good/fair (yellow); mediocre (orange); negligible (dark orange); bad (red). + = very low activity; ++ = low activity; +++ = high activity; ++++ = very high activity. Nd = not determined.

	Somatostatin		Somatostatin analogue (SSA)		
	SST-14	SST-28	ITF2984	Pasireotide	Octreotide
Receptor-binding mean IC_50_ (nM) (95% confidence intervals)					
SSTR1	nd	0.99 (0.79–1.26)	21.7 (14.2–33.2)	43.9 (31.8–60.4)	nd
SSTR2	nd	0.19 (0.064–0.40)	1.8 (1.35–2.5)	5.5 (4.58–6.66)	0.24 (0.20–0.28)
SSTR3	nd	0.22 (0.19–0.25)	0.35 (0.30–0.42)	2.73 (2.24–3.34)	3.25 (2.61–4.06)
SSTR5	nd	0.10 (0.056–0.18)	0.36 (0.29–0.46)	0.37 (0.28–0.49)	5.03 (3.69–6.85)
Receptor internalization in U2OS-transfected cells at 10^−5^ M (10^−6^ M)					
SSTR2	nd	51.4	37.3 (35.9)	49.2 (50.5)	74.1 (81.0)
SSTR3	nd	64.0	61.1 (62.5)	41.6 (46.3)	43.4 (55.2)
SSTR5	nd	84.5	34.6 (42.0)	40.5 (43.9)	34.6 (39.3)
Phosphorylation SSTR sites (HEK293-transfected)					
SSTR2—PS341/S343	++++	nd	++	+++	++++
SSTR2—PT535/T534	++++	nd			++++
SSTR2—PT536/T539	++++	nd			++++
SSTR3—PS337/T341	++++	nd	++++	++	+
SSTR3—PT438	++++	nd	++++	++	+
SSTR5—pT333	++++	nd	nd	++	+
SSTR-G protein signaling—GIRK2 channel activation in transfected HEK293 cell line (EC_50_ nM mean ± SEM)					
SSTR2	0.4 (±0.03)	nd	311.6 (±29.3)	60.0 (±16.3)	4.0 (±0.03)
SSTR3	0.8 (±0.1)	nd	10.7 (±1.7)	63.0 (±7.5)	51.9 (±10.0)
SSTR5	0.5 (±0.1)	nd	95.1 (±15.2)	16.5 (±2.5)	54.4 (±6.6)
SSTR-G protein signaling—GIRK2 channel activation in AtT-20 cell line					
WT	nd	nd	−8.125	nd	−7.5
Transfected SSTR3	nd	nd	−8	nd	−7.75
Other characteristics					
Half-time life	2.5 min	2.0 min	18 h	18 h	2 h
GH release (inhibition ≅ 1 nM)	nd	nd	50%	75%	100%

**Table 2 cancers-15-03453-t002:** Percentage of inhibition of opioid receptors induced by ITF2984 and Pasireotide at single doses. * Data from FDA Center for Drug Evaluation and Research—Application Number: 200677Orig1s000 PHARMACOLOGY REVIEW(S). Mean of results obtained with two different batches of Pasireotide.

Opioid Receptor	ITF2984	Pasireotide *
δ (DOP) (h) (agonist)	79	80
κ (KOP) (agonist)	97	109
µ (MOP) (h) (agonist)	97	90
NOP (ORL1) (h) (agonist)	61	-

## Data Availability

Not applicable.

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
