# Peer review of "Identification of a Novel SSTR3 Full Agonist for the Treatment of Nonfunctioning Pituitary Adenomas"

_cancers, 2023, doi:10.3390/cancers15133453_

Round 1

Reviewer 1 Report

The authors show in this manuscript the promising role of ITF2984, a somatostatin analog with full agonist activity in SSTR3, the main SSTR in non-functioning pituitary adenomas (NPPAs). Based on in vitro and in vivo works, the data seems to favor that ITF2984 may possibly represent a potential novel therapeutic option for patients with NFPAs.

This study is relevant, and certainly the identification of a potential medical therapy for NFPAs is of utmost importance, as like the authors commented, at present there is no good medical therapy options for the management of patients with NFPAs.

One of the shortcomings with the in vitro studies is that all work has been performed in cell lines, and it would have been extremely enriching and relevant if the authors would have conducted some work on primary cultures of non-functioning pituitary adenomas/gonadotrophinomas. Perhaps this can be done in the future?

Other major comments/aspects:

- The aim of this study is not clearly stated in the Introduction section, neither in the abstract.

- In the methods section, regarding RT-qPCR, lines 204-205: it is stated “specific sets of primers for these genes have been previously validated and reported.”, however no reference is provided. It should be added here a reference of a paper with the sequences of the primers used in the study, or otherwise, the authors should provide a list with all primers used in the study.

- Figure 1: There is a mistake in the label of Lanreotide. It should be “D”, not “E”. Still about Figure 1 a brief legend could be added to explain the main aspects of the structure of these analogues and the main differences among each other; otherwise, this figure will be useless and meaningless for many readers.

- Figure 4: please revise the abbreviations used in the western blot figure (on the left side). “SS-14” should be “SST-14” (as it is in the subtitle)? Also, SST3, SST2 and SST5 should be amended to SSTR3, SSTR2 and SSTR5 to ensure consistency with the subtitle and with the general text.

Minor aspects:

- Issue with how the references 1 and 2 appear in the first paragraph of the introduction (line 52). It should be displayed as “[1,2]” and not as “[1],[2]”

- Introduction, fourth paragraph: it is mentioned that “some studies” showed efficacy for the use of DRD2 agonist in NFPAs, but only one reference concerning only one study (not some) was provided. Please add here other studies showing this efficacy, or rephrase the sentence.

- Introduction section, lines 109-111: “Indeed, ITF2984 suppressed tu-109 mor growth in female, but not in male, rats having higher baseline expression of the Sstr3 110 gene versus males” - the last part of this sentence does not make sense to me. Can you please revise it?

- Legend of Figure 3: there is one “full stop” added as track changes, and still not accepted. Please correct this. Also, in the figure itself, in the boxes in the side it reads “SST2” and “SST3” – to my understanding it should read “SSTR2” and “SSTR3” – if so, please amend this.

- Results section, 3.5.2. Effect of ITF2984 on NFPAs proliferation rates: in this section the proliferation index is written in both ways, Ki67 and Ki-67. Please choose one, and use it consistently.

Author Response

Response to Reviewer 1

  1. We have indeed considered to do experiments on primary cultures, as suggested by the referee. Those cultures were not available to us so far but we have recently established a collaboration with a clinical group that is setting them up and we are planning to test ITF2984 as soon as samples will be available.
  2. The aim of the study is now better stated in the abstract (lines 35-36) and in the introduction (lines 105-106)
  3. The reference to the primers was added 
  4. Figure 1 was re-labelled and a more detailed figure legend was added.
  5. Figure 4: the abbreviations were corrected
  6. The reference citations in the first paragraph were corrected.
  7. The sentence in the fourth paragraph of the introduction was rephrased
  8. Lines 109-111 were re-written to improve the clarity, as suggested by the reviewer.
  9. Figure 3 was corrected in both legend and labels.
  10. The KI67 proliferation index is now written in a consistent way throughout the paragraph

Reviewer 2 Report

This is an original study aiming to characterize ITF2984, a novel pan SSTR agonist (structurally analogous to other SSTR agonists, but with a  higher affinity for SSTR3)

Unlike Pasireotide or Octreotide, ITF2984 induced SSTR3 internalization and phosphorylation and exhibited antitumoral effects in vivo in a strain of mutant rats that represent spontaneous, endogenous NFPA model

The study is very well designed, the results are clear and the discussion pertinent and well –written.

Overall the study is the result of a large amount of work,  the results are very significant and suggest a potential future use of ITF2984 in NFPA  patients or in other tumors/diseases where the SSR3 expression is prominent and active.

Author Response

We thank the reviewer for his comments